Original research

# Defining 'therapeutic value' of medicines: a scoping review

Camille E G Glaus, Andrina Kloeti, Kerstin N Vokinger 

Academic Chair for Regulation in Law, Medicine, and Technology, Faculty of Law and Faculty of Medicine, University of Zurich, Zurich, Switzerland

**Correspondence to**
Professor Kerstin N Vokinger;
Lst.vokinger@rwi.uzh.ch

## ABSTRACT

**Objectives** In recent years, discussions on the importance and scope of therapeutic value of new medicines have intensified, stimulated by the increase of prices and number of medicines entering the market. This study aims to perform a scoping review identifying factors contributing to the definition of the therapeutic value of medicines.

**Design** Scoping review.

**Data sources** We searched the MEDLINE, CINAHL, Embase, Business Source Premier, EconLit, Regional Business News, Cochrane, Web of Science, Scope and Pool databases through December 2020 in English, German, French, Italian and Spanish.

**Eligibility criteria** Studies that included determinants for the definition of therapeutic value of medicines were included.

**Data extraction and synthesis** Data were extracted using the mentioned data sources. Two reviewers independently screened and analysed the articles. Data were analysed from April 2021 to May 2022.

**Results** Of the 1883 studies screened, 51 were selected and the identified factors contributing to the definition of therapeutic value of medicines were classified in three categories: patient perspective, public health perspective and socioeconomic perspective. More than three-quarters of the included studies were published after 2014, with the majority of the studies focusing on either cancer disorders (14 of 51, 27.5%) or rare diseases (11 of 51, 21.6%). Frequently mentioned determinants for value were quality of life, therapeutic alternatives and side effects (all patient perspective), prevalence/incidence and clinical endpoints (all public health perspective), and costs (socioeconomic perspective).

**Conclusions** Multiple determinants have been developed to define the therapeutic value of medicines, most of them focusing on cancer disorders and rare diseases. Considering the relevance of value of medicines to guide patients and physicians in decision-making as well as policymakers in resource allocation decisions, a development of evidence-based factors for the definition of therapeutic value of medicines is needed across all therapeutic areas.

## STRENGTHS AND LIMITATIONS OF THIS STUDY

⇒ This scoping review was based on a comprehensive search, and articles were screened and analysed by two independent authors.
⇒ Potential studies could be missed due to the heterogeneity of terms.
⇒ Another categorisation of value determinants is possible outside of therapeutics.

## INTRODUCTION

In recent years, discussions on the importance of value of new medicines have intensified, stimulated by the increase of prices and number of medicines entering the market.[1] Several countries with price regulations (for example, Germany or France) have tasked health technology agencies to assess the added therapeutic value of new medicines as a basis for price negotiation.[2] Also, medical associations developed frameworks that enable the value assessment of medicines with the goal to support physicians and patients in their decision-making. For example, the European Society for Medical Oncology (ESMO) created the ESMO-magnitude of clinical benefit scale, and the American Society of Clinical Oncology (ASCO) implemented the ASCO-value framework specifically for cancer medicines.[3 4] Additionally, non-profit organisations such as the Institute for Clinical and Economic Review emerged with the goal of conducting and disseminating comparative effectiveness evaluations to, among other things, encourage fair pricing.[5]

Studies indicate that only a fraction of new medicines provide high added therapeutic value when applying the therapeutic rating of health technology agencies or medical associations.[6–8] However, value of medicines can vary depending on the underlying understanding of value.[9 10]

The importance of the determinants contributing to the assessment of therapeutic value of medicines will further increase in the upcoming years given the growth of high costs of medicines that threaten healthcare budgets across countries. To support ongoing discussions on how to define value, we undertook a scoping review of the literature to identify the determinants used to assess the therapeutic value of medicines.

## METHODS

### Selection criteria

This scoping review was conducted using the Preferred Reporting Items for Systematic Reviews and Meta-Analyses guidelines for scoping reviews. In December 2020, the databases MEDLINE (PubMed), CINAHL, Embase, Business Source Premier, EconLit, Regional Business News, Cochrane, Web of Science, Scope and Pool were searched for studies published until December 2020 in English, German, French, Italian and Spanish.

### Search strategy

The search string contained (value or worth) or ("value based") or (clinical* or medical* or therap* or cur* or pay* or drug* or pharmaceutical* or evaluat* or assess* or defin*). An additional operator was used so that there was a maximum of three words between the above terms and (valu* or worth). The second search string was (clinical* or medical* or therap* or cur* or patient*) and had an adjacent operator so that there was a maximum of three words benefit. The third search string was the following: ((drug* OR pharmac* OR medicine* OR medicat*) NEAR/20 pric*) OR AB= ((drug* OR pharmac* OR medicine* OR medicat*) NEAR/3 pric)). A more detailed explanation of the search strategy can be found in the online supplemental file.

### Screening protocol

Two authors (CEGG, AK) independently screened the articles following a fixed protocol. They both screened all abstracts and titles based on the inclusion criteria. They then compared their results and established a list with the non-congruent results. Disagreement was resolved in discussion with the last author (KNV). The same procedure was followed for the screening of the full texts (table 1).

### Study selection

For studies that appeared from their title or abstract to discuss determinants of the therapeutic value of medicines, full articles related to these titles were obtained and screened. Studies were included if they proposed a definition and/or one or more determinants of therapeutic value of medicines. We excluded studies that provided information about the value of specific medicines but did not include specific information on determinants of value. We also excluded studies that focused on related topics but did not directly address the determinants of therapeutic value, such as value-based pricing.

### Analysis

The first author (CEGG) reassessed the identified results and divided them in three categories: patient perspective, public health perspective and socioeconomic perspective. This decision was discussed with the last author (KNV). Studies that matched more than one category were included separately in each category. The studies were assessed from April 2021 to May 2022. This analysis was descriptive, neither the veracity nor the justifiability of the results was assessed.

The study adheres to the Enhancing the QUAlity and Transparency Of Health Research Reporting Guidelines. Detailed methods are presented in the online supplemental file.

### Patient and public involvement

None.

## RESULTS

The search strategy generated 1833 records, of which we identified 442 for full review. Forty-four unique papers were included. Reference mining led us to 7 additional articles that met our criteria, resulting in a total of 51 articles for the final sample. Figure 1 provides a flow diagram of the search results.

The dates of the articles range between 1996 and 2020, although fewer than one-quarter (11 of 50, 22%) were published prior to 2014.

Of the 51 studies, most determinants of value focused on cancer medicines (14 of 51, 27.5%) or medicines targeting orphan diseases (11 of 51, 21.6%). Five of 51 (9.8%) were for other therapeutic areas (such as chronic

**Table 1** Criteria for study selection

| Inclusion | Exclusion |
|---|---|
| ► Terms 'value' and 'benefit' if determinants of value of medicine or as argument for pricing | ► No direct link to 'value', 'HTA' and 'benefit' (eg, accessibility, innovation) |
| ► New tool/concept or proposed adoption of already existing tool/concept | ► Application of already existing tools/concepts |
| ► Tool/concepts by countries/state institutions if these countries/state institutions developed/adapted them or if they are unclear | ► Tool/concept by countries/state institutions if they only apply an already existing tool/concept |
| ► Journals | ► Complementary medicines |
| ► Conference papers | ► Biosimilars |
| | ► Generics |
| | ► Drugs (eg, randomised clinical trials) |
| | ► Contracting and insurance design |
| | ► News articles |

HTA, Health Technology Assessment.

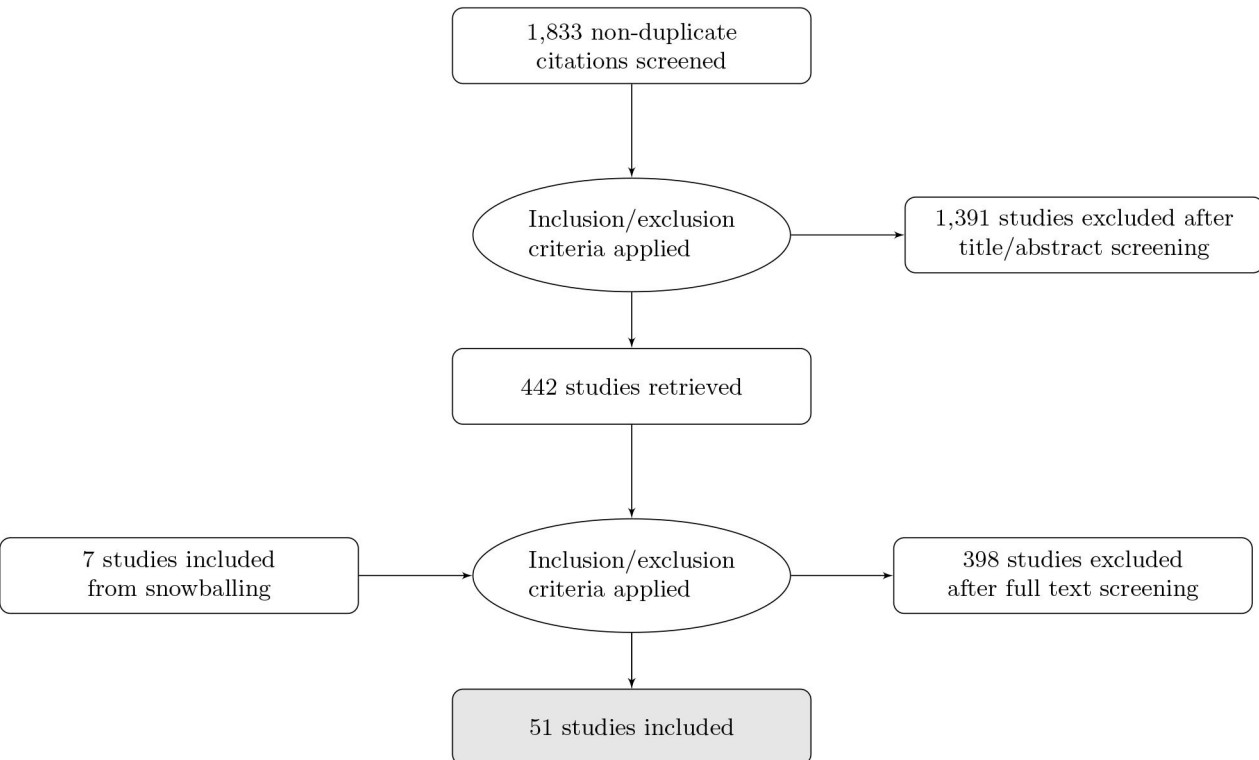

**Figure 1** Flow diagram of search results.

diseases[11] or neurological diseases[12]), and 21 of 51 (41.2%) were unspecified.

Some studies outlined a holistic approach for the assessment of value of medicines. For example, Schnipper and colleagues outlined that value is generally accepted as a measure of outcomes achieved per monetary expenditure.[3] However, in most cases, the studies did not incorporate a holistic approach, but rather focused on a perspective (eg, patient perspective) and on specific determinants (tables 2–4). We

| Table 2 | Identified determinants for the assessment of value of medicines from the patient perspective | | | |
|---|---|---|---|---|
| **Patient perspective** | | | | |
| **Category** | **Medical specialty** | | | |
| | **Oncology (N=14)** | **Orphan disease (N=11)** | **Other (N=5)** | **Not specified (N=21)** |
| Quality of life | 3 4 13 15 16 21 23 27 29 31 | 17 26 35 37 | 11 12 18 30 | 14 19 20 22 24 25 29 32–34 36 38 39 |
| Burden of disease | | | | |
| General definition | 29 41 42 | 17 37 43 44 46 47 | 30 | 19 24 32 36 49 54 |
| Severity of disease, symptoms | 3 13 15 16 21 27 31 40 41 | 17 26 35 37 43–47 50 | 11 30 48 | 24 25 28 32 34 36 39 49 51 52 54 |
| Unmet medical need, alternatives | 15 21 29 40–42 | 26 35 37 43 46 47 | 30 | 19 20 22 24 28 32 49 52–54 |
| Others: patient experience, fear of contagion, daily activity, reaching a landmark life event, dignity | 3 16 21 27 29 31 42 | 37 | 11 | 19 20 25 36 38 49 53 |
| Convenience | | | | |
| Posology, route of administration, instructions, comfort | 3 13 15 29 31 | 35 | 11 12 | 22 24 28 36 49 52 53 55 |
| The numbers listed in the medical specialty match the references. Studies that address multiple categories for the definition of value of medicines were listed for each category separately. | | | | |

**Table 3** Identified determinants for the assessment of value of medicines from the public health perspective

| Public health perspective | | | | |
|---|---|---|---|---|
| **Category** | **Medical specialty** | | | |
| | **Oncology (N=14)** | **Orphan disease (N=11)** | **Other (N=5)** | **Not specified (N=21)** |
| Epidemiological endpoints | | | | |
| Prevalence, incidence, rarity | 15 40 42 57 | 26 35 37 45 46 50 | | 24 |
| Clinical endpoints and evidence | | | | |
| Safety | 3 16 29 31 | 35 44 | 12 48 | 19 20 22 24 28 32 33 53 55 |
| Efficacy | 13 15 16 29 | 35 44 50 | | 19 20 22 24 32 33 52–55 |
| Effectiveness | 3 13 15 31 40 42 | 26 35 37 | 18 30 48 | 14 20 25 33 51 53 59 60 |
| Clinical benefits | 3 4 16 23 27 29 31 41 | 17 26 35 45–47 58 | 30 | 28 39 49 51 54 59 |
| Side effects, toxicity | 3 4 13 15 16 21 23 27 31 | 37 | 30 48 | 22 24 25 28 33 36 39 |
| Compliance, discontinuation, tolerability | 13 29 | 35 | 30 | 20 22 25 28 32 33 42 53 54 |
| Certainty, evidence | 3 16 31 42 57 | 26 37 47 50 58 | | 19 22 24 25 32 36 39 51 52 60 |

The numbers listed in the medical specialty match the references. Studies that address multiple categories for the definition of value of medicines were listed for each category separately.

divided the different determinants of value in three categories. Some determinants can be classified in more than one category. For example, we classified the determinant 'effectiveness' under the public policy perspective; alternatively, it could also fall into the patient perspective.

### Value of medicines: patient perspective

We identified 48 articles that matched this category, focusing on the following aspects: quality of life (31 studies),[3 4 11–39] burden of disease (including severity of the disease, unmet medical need, 39 studies)[3 11 13 15–17 19–22 24–32 34–37 39–54] and convenience (eg,

**Table 4** Identified determinants for the assessment of value of medicines from the socioeconomic perspective

| Socioeconomic perspective | | | | |
|---|---|---|---|---|
| **Category** | **Medical specialty** | | | |
| | **Oncology (N=14)** | **Orphan disease (N=11)** | **Other (N=5)** | **Not specified (N=21)** |
| Economic burden | | | | |
| Treatment costs | 3 15 29 31 | 44 | 11 12 | 22 24 33 34 36 |
| Non-treatment costs | 3 15 16 29 31 42 | 44 | 11 | 14 22 24 25 32 34 36 49 60 |
| Willingness to pay | 15 | | 11 30 | 14 28 32 33 51 |
| Innovation | | | | |
| Innovation | 40 41 | 17 44 | 11 | 22 24 25 32 36 49 52 53 |
| Spillover in research | 13 41 | 45 | | 24 34 36 49 59 60 |
| Mechanism of action, effectiveness, route of administration | 13 42 | | 48 | 19 24 51 52 54 55 |
| Others: research undertaken, unique indication, complexity of innovation, technological considerations | 41 | 26 35 43 | | 33 54 59 |
| Broader social impact | | | | |
| Social value | | | | 28 33 36 49 59 |
| Altruism | | | | 59 |
| Discrimination | | 17 | | 49 |
| Equity, fairness | 3 31 40 42 | 37 58 | 48 | 34 38 |
| Family benefit | 27 | 37 | | 25 28 59 |
| Others: prevention, risk reduction, feasibility (adoption), public health benefits | 15 16 | 45 50 | | 24 25 38 53 54 60 |

The numbers listed in the medical specialty match the references. Studies that address multiple categories for the definition of value of medicines were listed for each category separately.

administration route, 16 studies)[3 11–13 15 22 24 28 29 31 35 36 49 52 53 55] (table 2).

## Quality of life

Many studies acknowledged the importance of quality of life when assessing the value of medicines.[28 36] However, most studies did not offer a clear or unified definition for this determinant. Some studies specified quality of life as the impact of the treatment on the physical and mental abilities.[3 30 31 35] Another study highlighted that quality of life for patients with cancer should be assessed in a questionnaire that addresses four categories: social well-being, emotional well-being, memory and need of a caregiver.[27] A further study emphasised that quality of life is inherent to the characteristics of the individual patient.[11]

## Burden of disease

A unified definition could also not be identified for the value determinant burden of disease. For example, different definitions were applied for the determinant 'severity'. One study stated that 'severity of the disease relates to the condition's degree of seriousness in response to mortality and morbidity-derived disability […] or the expected remaining life years adjusted for their quality of life'.[56] Another study defined severity more broadly as the 'overall impact of a problem on an individual'.[25] This was specified in other studies, which stated that the severity of a disease included physical and mental aspects of the disease.[25 26 35]

Unmet medical need was a further frequently mentioned determinant for the evaluation of value of medicines, especially in the context of orphan diseases.[26 35 37 43 46 47] Unmet medical need was often defined as the therapeutic option for the patient or the number of available medicines for the specific disease.[21 26 35 41 42 47] Some studies underlined that the benefits of alternative medicines should also be considered.[35 47] Medicines targeting a disease with no alternative treatment option available should be granted a higher value.[47]

## Value of medicines: public health perspective

Forty-nine studies in our study cohort focused on this perspective when assessing value of medicines. We identified the following clusters: epidemiological endpoints (eg, size of population, 13 studies)[14 15 24 26 35 37 40 42 45 46 50 57 58] and clinical endpoints (eg, safety, efficacy or evidence, 45 studies)[3 4 12 13 15–25 27–33 35–37 39–42 44–55 57–60] (table 3).

## Epidemiological endpoints

Multiple studies referred to epidemiological endpoints (for example, incidence or prevalence), as an important aspect for the evaluation of value of medicine (10 studies).[15 24 26 35 37 42 45 46 50 57] Some studies argued that the rarity of a disease increases the value of the medicines targeting those diseases,[25 26 35] while others stated that this approach is unethical.[24 58] Some studies also argued that taking into account epidemiological endpoints would be unethical since that would unjustifiably increase the treatment value of certain diseases compared with others.[24 58]

## Clinical endpoints

Many studies considered outcomes of clinical trials, such as safety or effectiveness as core determinants for the evaluation of value of medicines,[3 4 12–20 22–33 35 37 39–42 44–55 58–60] even more so in more recent studies.[16 22 24 32] Trial outcomes are influenced by the understanding of the disease,[37] the stage of the disease (the benefit of medicines targeting the treatment of advanced-stage diseases may be more modest compared with those applied in a curative setting),[3 16 31 38] the duration of the study (some medicines result in a short tumour response and in case of short study duration, the long-term effects may be unclear),[16] the size of the targeted population evaluated,[26 37 47 50] the study design,[24 26] the subjects of the study and the potential resulting heterogeneity in medicine response.[11 16 22 27 32 60] Concerns with regard to the lack of evidence were particularly raised for medicines targeting orphan diseases and cancer medicines.[26 37 47 50] Value of medicines based on outcomes of clinical trials may change over a medicine's lifecycle depending on, among other things, new evidence provided in further trials,[16 32 58] additional indications that increase the aggregated value to society[13 24 59] and outcomes in long-term studies.[16]

Side effects and toxicity were other important determinants frequently highlighted as determinants for the assessment of value of medicines.[3 4 15 16 21–25 27 28 30 31 33 36 37 39 48 50 59] Multiple studies specified side effects and toxicity as tolerability, discontinuation or complications.[12 13 21 21 24 35] In the same context, several authors stressed that compliance with a medicine is an important element of value.[13 20 22 25 28–30 32 33 35 42 53 54]

## Value of medicines: socioeconomic perspective

We included 46 studies in the study cohort. We identified the following clusters: economic burden (eg, treatment costs, non-treatment costs and willingness to pay, 21 studies);[3 11–16 22 24 25 28 29 31–35 44 49 51 60] innovation (eg, mechanism of action or spillover effect, latter includes, for example, benefits that may evolve due to the treatment of the disease, 26 studies)[11 13 17 19 22 24–26 32 34–36 40–45 49 51–55 59 60] and broader social impact (eg, equity, 23 studies)[3 15–17 24 27 28 31 33 34 36–38 42 45 49 50 53–55 58–60] (table 4).

## Economic burden

Treatment costs and non-treatment costs (eg, caregiver costs, lost work productivity) were considered by multiple studies as important determinants to appraise value of medicines.[3 11–16 22 24 25 29 31–34 36 42 44 49 60] Non-treatment costs include also the financial burden for society, which was identified as a relevant determinant for the assessment of value of medicines.[11 24 33 37 38 49 50]

Some studies defined value as the willingness to pay for a medicine.[11 14 15 28 30 32 33 51] One study underlined the subjectivity of this criteria, highlighting that, for example, patients may have another willingness compared with physicians, and that the willingness to pay can also vary among patients or physicians.[11]

## Innovation

Many studies argued that innovation is another core element when assessing value of medicines.[11 13 17 19 22 24–26 32 34–36 40–45 49 51–55 59 60] There was not one unified definition for innovation. Examples included the medicine's novel mechanism of action,[13 19 24 42 51 52 54 55] its effectiveness and side effects,[48] its spillover effect (understood as the gained knowledge during the development of the medicine that serves as a basis for the development of further medicines)[13 24 34 36 41 45 59 60] or the route of medicine administration.[13]

Other identified determinants for the evaluation of value of medicines were the research and development costs,[26 33 35] or the level of complexity of the medicine.[26 35 43] Another study explicitly stated that research and development costs should not be considered for the assessment of value, arguing that value should only reflect the direct benefits for the patient.[14]

## Social impact

Some studies considered the broader social impact of medicines as important elements for the assessment of value. It includes general ethical principles, such as altruism towards the poor,[59] discrimination,[17 49] fairness[40] or equity.[3 31 34 37 38 42 48 58] It also comprises other societal determinants such as prevention,[24 25 38] and the relief of family members of emotionally and physically demanding responsibilities.[25 27 28 37 59]

## DISCUSSION

More than three-quarters of the included studies were published after 2014, with the majority of the studies focusing on either cancers or rare diseases. A major amount of the included studies highlighted only specific aspects of value of medicines that we classified in three categories: patient perspective, public health perspective and socioeconomic perspective.

Our study findings have implications for the ongoing discussions around value of medicines. Value assessment of medicines supports, for example, patients and physicians in decision-making in selecting the best treatment for an individual patient, or health technology agencies and policymakers in resource allocation and reimbursement decisions.[3 61] Given the importance of value assessment of medicines, our study findings suggest that more analysis and discussions are indicated to develop an evidence-based definition for and understanding of value. The identified determinants in this study for the value of medicines could serve as a basis in this regard. Frequently mentioned determinants for value were quality of life, therapeutic alternatives and side effects (all patient perspective), prevalence/incidence and clinical endpoints (all public health perspective), and costs (socioeconomic perspective). The specific determinants for the value assessment of medicines may vary depending on the country, healthcare system, therapeutic area and patient population.[3]

Value determinants, such as a medicine's evidence, clinical outcomes or its impact on quality of life can change over time based either on new findings from additional clinical trials or real-world evidence or in comparison with new medicines entering the market for the same indication.[59] These study findings suggest that the value of medicines should not be understood statically but rather dynamically, which is consistent with the approach of Health Technology Assessment (HTA) bodies or agencies that reassess value and prices of medicines either on a regular basis (eg, every 3 years in Switzerland) or if new evidence from clinical trials or real-world data is provided (eg, Germany).

Many of the studies in the study cohort were published in recent years, with the majority of the studies focusing on either cancers (14 of 51, 27.5%) or rare diseases (11 of 51, 21.6%). The focus on these medicines might be explained by their high treatment prices, their potentially severe adverse events and the increasing amount of medicines in these therapeutic areas entering the market.[62 63] Furthermore, medical associations such as ASCO and ESMO developed value frameworks specifically for cancer medicines. By contrast, for example, only one included study focused on the determinants of value specifically for diabetic medicines.[18] Also, other therapeutic areas, such as psychiatry or neurology, are not or only marginally addressed in the included studies. A stronger focus on the factors of value of medicines in general and for therapeutic areas other than cancer and rare diseases is indicated.

## Limitations

This study has limitations. Due to the broadness of the topic, it is possible that the study cohort is not exhaustive. Our multilayered search strategy allowed us to screen a wide range of studies, leaving us confident that we reached thematic saturation for main outcomes. However, we focused only on determinants defining the therapeutic value of medicines and excluded the rich literature on related topics such as 'value in healthcare' or 'healthcare services'. We excluded reports and protocols of HTA bodies, which may also provide important insights into the determinants of therapeutic value. This would be interesting to assess in a follow-up study; however, it would need the consultation of other databases compared with this literature search. Furthermore, we only focused on studies published in English, German, French, Italian or Spanish. Thus, it is possible that we did not include relevant studies published in other languages. Lastly, another categorisation of value determinants is possible. Some determinants fulfil more than one category. In such cases, we categorised the determinant in the category that we consider as more adequate. For example, 'quality of life' is not only relevant from a patient perspective, but is also a public health consideration.

## CONCLUSIONS

Multiple determinants have been developed to define the therapeutic value of medicines. Most definitions

and determinants identified for the assessment of therapeutic value of medicines were developed for cancer disorders and rare diseases. Considering the relevance of value of medicines to guide patients and physicians in decision-making as well as policymakers in resource allocation decisions, a development of evidence-based factors for the definition of therapeutic value of medicines is needed, addressing also diseases outside cancer disorders and orphan diseases.

**Contributors** CEGG contributed to the data collection, data analysis, data interpretation, first rudimentary draft and revision of the manuscript. AK contributed to the data collection and data analysis. KNV contributed to the acquisition of funding, conception, data collection, data interpretation, first draft and revision of the manuscript. CECG had primary responsibility for the overall content as the guarantor. All authors have access to all the data in the manuscript.

**Funding** This study was funded by the Swiss National Science Foundation (SNSF, grant number 194607).

**Competing interests** None declared.

**Patient and public involvement** Patients and/or the public were not involved in the design, or conduct, or reporting, or dissemination plans of this research.

**Patient consent for publication** Not required.

**Ethics approval** The study did not require institutional review board approval because the data were based on publicly available information.

**Provenance and peer review** Not commissioned; externally peer reviewed.

**Data availability statement** Data are available upon reasonable request.

**ORCID iD**
Kerstin N Vokinger http://orcid.org/0000-0002-6997-7384

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
