## [Reviewer comments · BMJ Open]

ARTICLE DETAILS

TITLE (PROVISIONAL)	Defining 'therapeutic value' of medicines: a scoping review
AUTHORS	Glaus, Camille; Kloeti, Andrina; Vokinger, Kerstin

VERSION 1 – REVIEW

REVIEWER	Cherny, Nathan Shaare Zedek Medical Center, Department of Oncology, Cancer Pain and Palliative Medicine Service
REVIEW RETURNED	28-Dec-2022

GENERAL COMMENTS	The subject addressed by this paper is an interesting and important one. In this paper the authors have reviewed and taxonomised of the factors that contribute to decisions about the value of new therapeutic interventions. I have four concerns about this paper both of which are relatively minor and which could be easily addressed by the authors: 1) relates to the stated aim of the paper. The authors write "we are a systematic review of the literature of the taxonomy for the definition of value in medicines". I am concerned that this is an accurate description of the past that they have undertaken. Reading the manuscript was apparent is that this is not so much the definition of value but rather a scoping review of the factors contributing to the evaluation of value of a new therapeutic indication. The reframing of the aim in no way diminishes from the importance of this contribution that it more accurately describes what has actually been done. 2) relates to the presentation of findings: The findings are presented into three subsections: patient perspective, public health perspective and socio-economic perspective. In each of these and number of variables are described in free text. I think it would be value in having the different variables highlighted in further subheadings. For instance in the section on patient perspective this should be separate subheadings for quality of life, , burden of disease, convenience and unmet need. The section on socio-economic perspective should be subheadings of treatment and costs, willingness to pay, innovation, research and development costs, general ethical principles, other societal determinants. 3) taxonomy is somewhat siloed
---

	The presents some factors as the exclusive domain of either patient or public health whereas it is probably true for both. This particularly pertains to considerations of safety and effectiveness and side-effects. I find it absolutely implausible that these are not important considerations are held by patients or reflected in papers describing the attitude of patients towards value. 4) Regarding limitations I think that it is important for the authors to add that these contributing factors have been presented descriptively and not analytically insofar as the authors have not made any formal evaluation is to the veracity or justifiability one of individual factors regarding their contribution to the value assessment. This is particularly true for some of the considerations that they have highlighted including rarity and innovation both of which are somewhat contentious. Concluding comments In conclusion I do think that this paper would benefit from some reframing and restructuring but that it will make a meaningful contribution to future discussions regarding the spectrum of factors contributing to the evaluation of value from new
--	--

REVIEWER	Jenei, Kristina LSE, Health Policy
REVIEW RETURNED	05-Feb-2023

GENERAL COMMENTS	Thank you for the opportunity to review this manuscript. The question is relevant given different conceptions of value for pharmaceuticals. I have one overarching comment and a few smaller points. One overall comment about the focus of the manuscript. It is important to acknowledge that the concept of value has a long tradition in (health) economics. The authors specifically looked at definitions for medicines, which, they summarized into three categories. However, if the search would have expanded to "value in health care" or "health care services" (as opposed to specifically "medicines/pharmaceuticals") they would have likely found a larger, and richer body of work, complete with definitions and frameworks. This is often the case as the "medicines movement" (focus on medicines as a distinct aspect of health care) is newer. Given the authors found impartial definitions in the literature, the development of a definition for value in pharmaceuticals remains an outstanding gap that this manuscript has not addressed. However, the authors are specific in their aims to outline the literature, therefore this comment is likely out of scope for this specific manuscript. But I believe this point would be important to acknowledge in the discussion and/or limitations. -The authors amended the study protocol to concentrate on "therapeutic value" (and not value-based pricing, etc...). Given the large literature on value in economics, outlining the focus on therapeutic value in the methods/manuscript would reinforce the tight scope of this search to readers. - I am not sure if this is a systematic review or a scoping review. I may be wrong here, but it is my understanding that a systematic review involves a risk of bias assessment for the studies included in the review. According to the PRISMA checklist, this information is reported on page 8 (and figures 2-6). But I cannot find the details of how this was assessed. -Abstract: In the Conclusion: "...value of medicines is indicated." I am not sure what "indicated" means here. A definition is "needed"?
---

	-The discussion would benefit from some engagement about how/ or if the value of pharmaceuticals might differ from other therapeutic areas (or simply, how the definition might vary). https://academic.oup.com/intqhc/article/33/4/mzab140/6426034 https://www.nejm.org/doi/suppl/10.1056/NEJMp1011024/suppl_file/nejmp1011024_appendix1.pdf
--	---

VERSION 1 – AUTHOR RESPONSE

Reviewer 1 comments	Author response and changes made	Page number in revised paper
The authors write “we are a systematic review of the literature of the taxonomy for the definition of value in medicines”. I am concerned that this is an accurate description of the past that they have undertaken. Reading the manuscript was apparent is that this is not so much the definition of value but rather a scoping review of the factors contributing to the evaluation of value of a new therapeutic indication. The reframing of the aim in no way diminishes from the importance of this contribution that it more accurately describes what has actually been done	Thank you. We agree that a scoping review is more appropriate focusing on the different determinants relevant for the value assessment of new therapeutic indications is more relevant. We revised our study accordingly.	1-12
The findings are presented into three subsections: patient perspective, public health perspective and socio-economic perspective. In each of these and number of variables are described in free text. I think it would be value in having the different variables highlighted in further subheadings. For instance in the section on patient perspective this should be separate subheadings for quality of life, , burden of disease, convenience and unmet need. The section on socio-economic perspective should be subheadings of treatment and costs, willingness to pay, innovation, research and development costs, general ethical principles, other societal determinants.	Thank you for this comment. We agree that this provides a better structure of our manuscript. We adapted our text accordingly and added subheadings. We decided to subdivide the ‘Results’ section according to the main clusters identified (Tables 2, 3 and 4).	5-9
The presents some factors as the exclusive domain of either patient or public health whereas it is probably true for both. This particularly pertains to considerations of safety and effectiveness and side-effects. I find it absolutely implausible that these are not	We agree that our categorization is somewhat arbitrary as some elements - including safety, efficacy and adverse events - may fall into one category or the other. We decided to include these	5

important considerations are held by patients or reflected in papers describing the attitude of patients towards value	aspects in the public policy perspective because they are crucial aspects of therapeutic values for the authorities. We have clarified this in the section 'Results.' "Some determinants could arguably fall into one category or the other. For example, we classified the determinant 'effectiveness' under the public policy perspective, but it could also fall into the patient perspective."	
I think that it is important for the authors to add that these contributing factors have been presented descriptively and not analytically insofar as the authors have not made any formal evaluation as to the veracity or justifiability of individual factors regarding their contribution to the value assessment. This is particularly true for some of the considerations that they have highlighted including rarity and innovation both of which are somewhat contentious. In conclusion I do think that this paper would benefit from some reframing and restructuring but that it will make a meaningful contribution to future discussions regarding the spectrum of factors contributing to the evaluation of value from new	We clarified this in the section "Methods." "This analysis is descriptive; neither the veracity nor the justifiability of the results were assessed"	5

Reviewer 2 comments	Author response and changes made	Page number in revised paper
One overall comment about the focus of the manuscript. It is important to acknowledge that the concept of value has a long tradition in (health) economics. The authors specifically looked at definitions for medicines, which, they summarized into three categories. However, if the search would have expanded to "value in health care" or "health care services" (as opposed to specifically "medicines/pharmaceuticals") they would have likely found a larger, and richer body of work, complete with definitions and frameworks. This is often the case as the "medicines movement " (focus on medicines as a distinct aspect of health care) is newer. Given the authors found impartial definitions in the literature, the development of a definition for value in pharmaceuticals remains an outstanding gap that this manuscript has not addressed. However, the authors are specific in their aims to outline the literature, therefore this comment is likely out of scope for this specific manuscript. But I believe this point would be important to acknowledge in the discussion and/or limitations.	Thank you for this important comment. As you mentioned we restricted our analysis to medicines. As the editors suggested, we expanded our 'Methods' section to better underline the scope of our manuscript . We also acknowledged in the 'limitations' section that our study only focuses on value of medicines and thus excludes "value in health care" by outlining the following: "This study has	3-5 and 10

	limitations. Due to the broadness of the topic, it is possible that the study cohort is not exhaustive . Our multilayered search strategy allowed us to screen a wide range of studies, leaving us confident that we reached thematic saturation for main outcomes. However, we focused only on determinants of value of medicines and excluded the rich literature on related topics such as “value in healthcare” or “health care services”. Furthermore, we only focused on studies published	
--	---	--

	in English, German, French, Italian or Spanish. Thus, it is possible that we did not include relevant studies published in other languages. Lastly, another categorization of value determinants is possible.”	
The authors amended the study protocol to concentrate on "therapeutic value" (and not value-based pricing, etc...). Given the large literature on value in economics, outlining the focus on therapeutic value in the methods/manuscript would reinforce the tight scope of this search to readers.	Thank you for this input. We further clarified the scope of our research under the 'Method' section. “We also excluded studies that dealt with related topics but did not directly address the determinants of therapeutic value, such as value-	3-5

	based pricing.” As suggested by the Editors, we also included Table 1 in the main text.	
I am not sure if this is a systematic review or a scoping review. I may be wrong here, but it is my understanding that a systematic review involves a risk of bias assessment for the studies included in the review. According to the PRISMA checklist, this information is reported on page 8 (and figures 2-6). But I cannot find the details of how this was assessed.	We agree. Our study is indeed closer to a scoping review. We modified our manuscript accordingly.	2-10
Abstract: In the Conclusion: "...value of medicines is indicated." I am not sure what "indicated" means here. A definition is "needed"?	Thank you for this comment. We wanted to underline that there is a need for the development of an evidence-based definition of value. We rephrased the sentence to better express our opinion.	10
The discussion would benefit from some engagement about how/ or if the value of pharmaceuticals might differ from other therapeutic areas (or simply, how the definition might vary). https://academic.oup.com/intqhc/article/33/4/mzab140/6426034	We agree this is a valuable input. We modified our	

https://www.nejm.org/doi/suppl/10.1056/NEJMp1011024/suppl_file/nejmp1011024_appendix1.pdf	manuscript accordingl y.	
---	-----------------------------	--

VERSION 2 – REVIEW

REVIEWER	Jenei, Kristina LSE, Health Policy
REVIEW RETURNED	19-Aug-2023

GENERAL COMMENTS	I thank the authors for the revisions to the paper. It has greatly improved since the last draft. I agree with the reframing as a scoping review re: determinants of therapeutic value. Some minor comments:  -Strengths and limitations: I would specifically mention the focus on therapeutic value in the sentence "Another categorization of value determinants in possible" ...outside therapeutics, or "in health economics" -I agree with the changes to categorization proposed by the other reviewer and think this change improves the readability of the manuscript.
---

REVIEWER	Cherny, Nathan Shaare Zedek Medical Center, Department of Oncology, Cancer Pain and Palliative Medicine Service
REVIEW RETURNED	21-Sep-2023

GENERAL COMMENTS	This is a substantial improvement over the previous incarnation of this paper. It is much better framed as a scoping review of factor contributing to the evaluation of value. The choice to review the published scientific literature rather than the protocols and reports of HTA bodies is neither explained nor justified. In essence you have used a secondary source rather than a primary source . This should be clearly stated as a limitation of this review. You have taxonomized the factors contributing to value assessments according to as patient perspective, public health perspective or socioeconomic perspective. These categories are neither explained nor justified and, to this reader, they are not clearly justifiable. Furthermore the term perspective is somewhat misleading as it suggests that this is what patients are concerned about (which is patently untrue). If the terms are to be retained I suggest relabeling them as considerations rather than perspectives ie  1. Patient focused considerations 2. Public health focused considerations 3. Socioeconomic considerations
---

	I have other concerns about this taxonomy for instance  1. There is no clear justification why QoL is separated from “clinical endpoints”. Indeed QoL is one of the factors evaluated in as an important endpoint in clinical trials. 2. The section on QoL conflate QoL with other elements of the patient experience such as convenience and patient burden are not routinely evaluated in studies. 3. Since all of these issue are part of public health decision making, the separate category of public heath also does not work I am sorry but I don’t see the value in generating a new classification system of these factors that is not a substantial improvement on a more conventional classification such as:  1. Clinical effectiveness  1.1.1. Safety and efficacy 1.1.2. Comparative efficacy 1.1.3. Evidence quality 2. Economic considerations  2.1.1. Cost effectiveness 2.1.2. Total budget impact 2.1.3. Opportunity cost 3. Patient related considerations  3.1.1. Strength of indication 3.1.2. Convenience, burden 4. Indirect society benefits  4.1.1. Innovation 5. Ethical considerations  5.1.1. Rare diseases, orphan indications If you believe that there are advantages, then these need to be explained. If not I suggest that you recategorize the considerations along a more conventional taxonomic structure such as the one I have provided above. I hope that this feedback will assist in the further development of this work.
--	---

VERSION 2 – AUTHOR RESPONSE

Comments for Manuscript "Regulatory review duration and submission delay of drugs in the US and Europe, 2011-2020."

Reviewer 1	Author response and changes made	Page number in revised paper

I thank the authors for the revisions to the paper. It has greatly improved since the last draft. I agree with the reframing as a scoping review re: determinants of therapeutic value.	Thank you.	
Some minor comments: -Strengths and limitations: I would specifically mention the focus on therapeutic value in the sentence "Another categorization of value determinants in possible" ...outside therapeutics, or "in health economics"	Thank you for this point, we modified this section accordingly.	3
I agree with the changes to categorization proposed by the other reviewer and think this change improves the readability of the manuscript.	Thank you for your comment, which we addressed.	

Reviewer 2	Author response and changes made	Page number in revised paper
This is a substantial improvement over the previous incarnation of this paper. It is much better framed as a scoping review of factor contributing to the evaluation of value.	Thank you.	
The choice to review the published scientific literature rather than the protocols and reports of HTA bodies is neither explained nor justified. In essence you have used a secondary source rather than a primary source. This should be clearly stated as a limitation of this review.	Thank you for this comment. We are aware that HTA bodies have published protocols that include determinants of therapeutic value. However, as outlined in the introduction and methods section, we are interested in how the scientific literature defines therapeutic value. It would be interesting to assess in a follow-up study how the definition of therapeutic value differs between HTA-bodies (which incorporates another search strategy than the approach in our study). To be clearer about our approach, we additionally clarified this also in	9

	the limitations section in the revised manuscript.	
You have taxonomized the factors contributing to value assessments according to as patient perspective, public health perspective or socioeconomic perspective. These categories are neither explained nor justified and, to this reader, they are not clearly justifiable. Furthermore the term perspective is somewhat misleading as it suggests that this is what patients are concerned about (which is patently untrue). If the terms are to be retained I suggest relabeling them as considerations rather than perspectives ie  1. Patient focused considerations 2. Public health focused considerations 3. Socioeconomic considerations I have other concerns about this taxonomy for instance  1. There is no clear justification why QoL is separated from “clinical endpoints”. Indeed QoL is one of the factors evaluated in as an important endpoint in clinical trials. 2. The section on QoL conflate QoL with other elements of the patient experience such as convenience and patient burden are not routinely evaluated in studies. 3. Since all of these issues are part of public health decision making, the separate category of public health also does not work I am sorry but I don’t see the value in generating a new classification system of these factors that is not a substantial improvement on a more conventional classification such as:  1. Clinical effectiveness  1.1.1. Safety and efficacy 1.1.2. Comparative efficacy 1.1.3. Evidence quality 2. Economic considerations  2.1.1. Cost effectiveness 2.1.2. Total budget impact 	Thank you for this valuable input. We have been discussed this taxonomy at length since initiation of our study and have tried different categorizations. We further realized that some aspects may fall in more than one category, regardless of how we define the categories. We agree that this constitutes a limitation, which we further emphasize in the revised limitations section. We appreciate your categorization. However, also here, some of the identified determinants can fall in more than one category – for example, quality of life can be a patient related consideration or an ethical consideration. If possible, we would prefer to keep our classification. We believe it is easy to understand and that it entails three major and important perspectives. We do not think that more categories are necessarily helpful. We are aware of its limitations and are transparent about them in the limitations section.	10

2.1.3. Opportunity cost 3. Patient related considerations 3.1.1. Strength of indication 3.1.2. Convenience, burden 4. Indirect society benefits 4.1.1. Innovation 5. Ethical considerations 5.1.1. Rare diseases, orphan indications If you believe that there are advantages, then these need to be explained. If not I suggest that you recategorize the considerations along a more conventional taxonomic structure such as the one I have provided above. I hope that this feedback will assist in the further development of this work		
---	--	--

1

VERSION 3 – REVIEW

REVIEWER	Cherny, Nathan Shaare Zedek Medical Center, Department of Oncology, Cancer Pain and Palliative Medicine Service
REVIEW RETURNED	17-Nov-2023

GENERAL COMMENTS	This is a good short review of the factors considered by health technology assessment bodies in considering the value of a medicine or technology. The authors use a simplified classification of the factors considered, that some readers may find helpful.
---